# OpenReview forum: "Unlearning Sensitive Information in Multimodal LLMs: Benchmark and Attack-Defense Evaluation"
_TMLR — Accepted by TMLR_

### Review · Reviewer_kEAH · 2024-09-15

**Summary Of Contributions:**

This paper introduces UnLOK-VQA, a novel multimodal unlearning benchmark designed to evaluate the deletion of specific multimodal knowledge from MLLMs. Building on this, the paper presents an extensive evaluation framework that includes attack and defense strategies to assess the effectiveness of various unlearning methods. It highlights that the latest editing methods like LoRA fine-tuning cannot fully delete knowledge from MLLMs.

**Audience:**

Yes

**Claims And Evidence:**

Yes

**Requested Changes:**

The introduction of RLHF-related work on the second page of the manuscript is necessary. Moreover, while the author refers to in-context relearning, there is an apparent lack of a comprehensive introduction and corresponding literature citations to support this concept.

**Strengths And Weaknesses:**

# Strengths

1.	This paper introduces a new dataset for evaluating the deletion of specific undesirable multimodal knowledge from models.

2.	It shows that the latest editing methods like LoRA fine-tuning cannot fully delete knowledge from MLLMs, which is insightful.

# Weaknesses

In Contribution 2, the author presents an important finding: state-of-the-art model editing methods like LoRA fine-tuning cannot fully delete knowledge from MLLMs. However, there are several issues with this finding:

1.	Comparison with Full-Parameter Fine-Tuning: The comparison between LoRA and full-parameter fine-tuning is missing. Given that LoRA only updates a small number of model parameters, it remains unclear whether full-parameter fine-tuning could delete knowledge from MLLMs.

2.	Ablation Study Analysis for LoRA: Would increasing the rank size $r$ potentially delete knowledge from MLLMs? A more in-depth analysis of the ablation studies on LoRA is needed.

3.	Other Fine-Tuning Methods: It should be clarified whether other methods for fine-tuning MLLMs, such as Adalora or Adapter, could achieve similar effects.

4.	The Neighborhood Image only includes "Easy" and "Hard," which differs from other settings. It is suggested that the authors explain the reason for this.

5.	Standards that should be further clarified in Section 3.4.

---

> ### Author Response · Authors · 2024-10-13
> **Response to Reviewer kEAH - Part 1**
>
> > Comparison with Full-Parameter Fine-Tuning: The comparison between LoRA and full-parameter fine-tuning is missing. Given that LoRA only updates a small number of model parameters, it remains unclear whether full-parameter fine-tuning could delete knowledge from MLLMs.
> - In our unlearning setup, we utilize LoRA because it enables us to control the extent of parameter changes by adjusting the rank and alpha values, allowing us to make minimal adjustments to avoid collateral damage. We focus on LoRA rather than full-parameter fine-tuning due to our objective of unlearning specific information, which limits the amount of data available for training many parameters.
> Our experiments evaluating the Fact Erasure defense show that full-parameter fine-tuning results in higher Delta accuracies on both random and neighborhood data as shown in the table below. However, LoRA fine-tuning proves advantageous for making more targeted edits while minimizing collateral effects, as it introduces fewer unintended changes to the model's performance on unrelated data. This highlights LoRA’s strengths in scenarios where precise control over knowledge updates is crucial.
>
> | HP      | PD    | PD2   | HP+FT | Hard Img Rephrase | Hard Ques Rephrase | MM    | Rand Delta-Acc | Q Neigh Delta-Acc | Img Neigh Delta-Acc | Rewrite score |
> |---------|-------|-------|-------|-------------------|--------------------|-------|----------------|-------------------|---------------------|---------------|
> | LoRA    | 0.300 | 0.296 | 0.264 | 0.412             | 0.320              | 0.390 | 0.455          | 0.001             | 0.008               | 0.059         | 0.956 |
> | Full FT | 0.021 | 0.035 | 0.027 | 0.043             | 0.103              | 0.167 | 0.218          | 0.239             | 0.232               | 0.152         | 1     |
>
>
>
>
> > Ablation Study Analysis for LoRA: Would increasing the rank size r potentially delete knowledge from MLLMs? A more in-depth analysis of the ablation studies on LoRA is needed.
> - Our experiments show that increasing r leads to collateral damage in the model, as evidenced by a rise in Delta-Accuracy, indicating more unintended changes across tasks. This suggests that while a larger r allows for greater flexibility in modifying the model's behavior, it also increases the likelihood of altering unintended areas of knowledge, making LoRA less precise in its edits.Moreover, varying r did improve robustness against adversarial attacks. However, the targeted nature of LoRA's updates is compromised.
> | r  | PD2   | MM    | Rand Delta-Acc | Q Neigh Delta-Acc | Img Neigh Delta-Acc | Rewrite score |
> |----|-------|-------|----------------|-------------------|---------------------|---------------|
> | 1  | 0.300 | 0.455 | 0.001          | 0.008             | 0.059               | 0.956         |
> | 8  | 0.251 | 0.397 | 0.053          | 0.078             | 0.074               | 1             |
> | 32 | 0.124 | 0.173 | 0.138          | 0.091             | 0.085               | 1             |
> | 64 | 0.037 | 0.138 | 0.201          | 0.153             | 0.127               | 1             |
>
>
>
>
>
>
> > Other Fine-Tuning Methods: It should be clarified whether other methods for fine-tuning MLLMs, such as Adalora or Adapter, could achieve similar effects.
> - To investigate whether alternative fine-tuning methods, such as AdaLoRA or Adapters, could achieve similar effects to LoRA in modifying MLLMs, we conducted experiments using AdaLoRA. We tune the hyperparameters in AdaLoRA to achieve a rewrite score> 95% while maintaining a random delta-acc of <5%. Our findings show that the observations and results remain consistent with those of LoRA. This is because both methods operate on a similar principle: they aim to make constrained, targeted edits by fine tuning model weights, focusing on deleting specific information while minimizing collateral damage to the model’s overall knowledge.
>
> | r       | PD2   | MM    | Rand Delta-Acc | Q Neigh Delta-Acc | Img Neigh Delta-Acc | Rewrite score |
> |---------|-------|-------|----------------|-------------------|---------------------|---------------|
> | LoRA    | 0.300 | 0.455 | 0.001          | 0.008             | 0.059               | 0.956         |
> | AdaLoRA | 0.371 | 0.487 | 0.008          | 0.013             | 0.71                | 0.963         |

---

> ### Author Response · Authors · 2024-10-13
> **Response to Reviewer kEAH - Part 2**
>
> > The Neighborhood Image only includes "Easy" and "Hard," which differs from other settings. It is suggested that the authors explain the reason for this.
> - This choice was made because we attempted an additional method to generate medium-level neighborhood images but found it ineffective in altering the image to fit an alternate answer. Specifically, we followed a similar process used to generate medium rephrase images, employing Grounded SAM to minimally repaint the original image so that the answer to the original question becomes the alternate answer a′. To guide Grounded SAM, we prompted Flan-T5-XXL to extract the subject from the question q leading to the answer a, and then modified the subject in the image to reflect the alternate answer a′.
> However, these images turned out to be the hardest to modify, as the majority of the pixels remained unchanged, making them lie within a smaller radius neighborhood compared to medium-level images. This approach did not effectively alter the image enough to change the answer, especially in cases involving multi-hop reasoning, such as samples from OK-VQA. As a result, we excluded this method from our study.
> This explanation is present in Appendix D.
>
> > Standards that should be further clarified in Section 3.4.
>
> **Standards for Human Evaluation**
>
> - In our human evaluation process for UnLOK-VQA, we established clear standards for annotators to assess the quality of the generated data. The evaluation focused on two primary criteria:
>   - Consistency of Target Answers: Annotators were tasked with determining whether the target answer remains consistent when evaluating rephrased data. A consistent answer indicates that the rephrase effectively captures the original intent of the question.
>   - Appropriateness of Answer Changes: For neighborhood data, evaluators assessed whether the target answer changes appropriately in response to the modified question or context. An appropriate change signifies that the alteration aligns with the expectations set by the original question.
> - When we say x% of the data meets our standards, we mean that for x% of the questions, the annotators' annotations match the expected answer.
> We have clarified this in the updated version of the paper.
>
> > The introduction of RLHF-related work on the second page of the manuscript is necessary. Moreover, while the author refers to in-context relearning, there is an apparent lack of a comprehensive introduction and corresponding literature citations to support this concept.
>
> - The current mainstream approach is RLHF (reinforcement learning from human feedback, and its variants) [1,2]. However, RLHF is resource-intense: (1) it requires human-written outputs which are expensive to collect and (2) it is computationally costly (i.e. the standard three-stage aligning procedure).
> - In-Context Relearning [3, 4] involves using various non-jailbreak prompting strategies to extract unlearned knowledge from models.
>
> We have added these references in the updated version of the paper.
>
> [1] Ouyang, Longx, et al. "Training language models to follow instructions with human feedback." Advances in neural information processing systems 35 (2022): 27730-27744.
>
> [2] Bai, Yuntao, et al. "Constitutional ai: Harmlessness from ai feedback." arXiv preprint arXiv:2212.08073 (2022).
>
> [3] Lynch, Aengus, et al. "Eight methods to evaluate robust unlearning in llms." arXiv preprint arXiv:2402.16835 (2024).
>
> [4] Shi, Weijia, et al. "Detecting pretraining data from large language models." arXiv preprint arXiv:2310.16789 (2023).

---

### Review · Reviewer_jwQq · 2024-09-24

**Summary Of Contributions:**

The authors proposed a pipeline to create VQA samples to evaluate the influence of an unlearned MLLM from the perspectives of generalization and specificity. Following the proposed pipeline, the authors constructed their UnLOK-VQA dataset based on an existing benchmark dataset OK-QVA, which is evaluated by several information extraction and deletion methods.

**Audience:**

Yes

**Broader Impact Concerns:**

I do not have any broader impact concerns.

**Claims And Evidence:**

Yes

**Requested Changes:**

Besides providing clarification responses to the above weaknesses concerns, the following requested changes are expected.

- The authors apply their pipeline only to the specific OK-VQA dataset. To validate the generalizability of the proposed method, more VQA datasets should be considered. Otherwise, a corresponding explanation of this decision is needed.

- In Adversary's Objective (Section 4.1), the authors should better explain why their setting is more general than typical one-shot settings, which could improve the comprehension of this work's contributions.

- In Section 6.2, only two LLaVA model sizes with 7B and 13B parameters are evaluated to draw conclusions on the influence of model size on vulnerabilities. Other model architectures with varying model sizes should be evaluated for a more convincing conclusion. If additional experiments are not possible, a relevant discussion is needed.

__Minor Comments:__

- Equation (2) could be introduced earlier, e.g., in Section 3.1 when it is mentioned.

- Tables 5 and 6 should be placed before the references.

**Strengths And Weaknesses:**

__Strengths:__

- The evaluation involves various information extraction and deletion methods.

- The pipeline is clearly illustrated, which is easy to understand.

__Weaknesses:__

- The challenge(s) that this work aims to solve has/have not been well explained. More details are expected as to why generalization and specificity are important in this task and why existing works do not support them.

- A mismatch exists between the authors' design and examples. Specifically, the design of medium-level rephrase images is to substitute one random object in the original image with an object generated by Grounded SAM (Section 3.2 and Figure 2). However, the examples provided in Table 7 show that medium-level rephrase images are created by removing a part of the image (instead of objects).

---

> ### Author Response · Authors · 2024-10-13
> **Response to Reviewer jwQq - Part 1**
>
> > The challenge(s) that this work aims to solve has/have not been well explained. More details are expected as to why generalization and specificity are important in this task and why existing works do not support them.
> Here’s an in-depth motivation for generalization and specificity in the context of multimodal knowledge deletion:
>
>
> **Generalization**
>
> - In knowledge deletion tasks, generalization refers to the ability of a deletion method to ensure that deleted information cannot be recovered, even when an adversary rephrases questions or uses similar but not identical images. This is important because:
> Adversarial Robustness: A strong deletion method should be robust not just to the exact question or image for which the deletion was performed, but also to variations. Models trained on large multimodal datasets often display a broad understanding of concepts and can generalize across inputs. Without addressing generalization, a deletion method would only be effective for very specific input patterns, leaving the system vulnerable to broader attacks.
>
>
> - Existing works often focus on deletion in unimodal (usually text-based) settings, where generalization might involve only text rephrasing. However, in multimodal models, the challenge is compounded because adversaries can attack from two modalities—text and images. Existing research lacks effective frameworks to test how well deletion methods generalize across these different multimodal inputs. In this work, we focus on the creation of a dataset that enables evaluation of generalization of the deletion method with the help of rephrase images and rephrase questions of varying difficulty levels (varying proximity to the deleted data point).
>
>
> **Specificity**
>
>
> Specificity is the ability to ensure that only the targeted information is deleted without damaging the model’s broader knowledge. It’s crucial because:
>
> - Collateral Damage: When deleting a specific piece of information from a model, there is a risk of inadvertently erasing other related knowledge. This can cause the model to become less accurate in tasks that require related but not identical knowledge.
>
> - Maintaining Model Utility: After the deletion, the model should still function well on questions or images that lie in the "neighborhood" of the deleted information. If the deletion process is too aggressive, the model may lose accuracy on tasks that require similar or related knowledge, thereby reducing its utility.
>
> In contrast to existing works—which may only focus on the deletion of single facts in unimodal contexts—this work introduces the concept of neighborhood data for multimodal inputs to test specificity. It assesses whether the deletion method can remove specific knowledge without negatively impacting nearby but unrelated knowledge in the model’s broader understanding. We focus on the creation of a dataset that enables evaluation of specificity of the deletion method with the help of neighborhood images and neighborhood questions of varying difficulty levels (varying proximity to the deleted data point).
>
> We have added this explanation in the updated version.
>
>
>
>
>
> > A mismatch exists between the authors' design and examples. Specifically, the design of medium-level rephrase images is to substitute one random object in the original image with an object generated by Grounded SAM (Section 3.2 and Figure 2). However, the examples provided in Table 7 show that medium-level rephrase images are created by removing a part of the image (instead of objects).
> - The confusion arises from how "replacing an object generated by Grounded SAM" is implemented. In this case, Grounded SAM repaints the part of the image corresponding to the removed object rather than replacing it with a new object. This process is explained in the last line of the medium rephrase image generation paragraph in Section 3.2. We have clarified this point in the updated version to avoid further misunderstandings.

---

> ### Author Response · Authors · 2024-10-13
> **Response to Reviewer jwQq - Part 2**
>
> > The authors apply their pipeline only to the specific OK-VQA dataset. To validate the generalizability of the proposed method, more VQA datasets should be considered. Otherwise, a corresponding explanation of this decision is needed.
> - We appreciate the reviewer’s suggestion to explore more VQA datasets to assess the generalizability of our method. However, our decision to focus on the OK-VQA dataset was intentional due to its strong emphasis on factual knowledge, which aligns with the goals of our proposed knowledge deletion framework.
> Focus on Factual Knowledge:
> - Our method is specifically designed to evaluate the deletion of factual information from multimodal models. OK-VQA, unlike other VQA datasets, primarily requires external, factual knowledge to answer questions, making it an ideal choice for testing the effectiveness of knowledge deletion. Extending the pipeline to VQA datasets that do not focus on factual knowledge would not align with the core objectives of our approach.
> - We had tried extending our pipeline to the usual VQAv2 dataset. However, these other VQA datasets contain predominantly Yes/No questions, where the concept of deletion of the answer is less meaningful for assessing factual knowledge removal. Therefore, extending the pipeline to such datasets did not serve the core goals of our approach.
>
>
> > In Adversary's Objective (Section 4.1), the authors should better explain why their setting is more general than typical one-shot settings, which could improve the comprehension of this work's contributions.
>
> - The typical one-shot settings in past works consider an extraction attack successful only when sensitive information is revealed with a single query (i.e., with an attack budget B = 1). However, this approach is overly restrictive and does not account for the range of possible real-world adversarial strategies. Our setting generalizes this by using the concept of an attack budget (B), allowing the adversary to generate a candidate set (C) of multiple possible answers.
> - This more general setting reflects plausible scenarios, where the adversary could either:
>    - Attempt multiple queries to guess sensitive information;
>    - Generate multiple potential candidates and act on any correct one;
>    - Verify the correctness of information, as in the case where the adversary is also the data owner trying to confirm that their sensitive data has been properly deleted.
>
> - In all these cases, the system is insecure if the correct answer is found among the candidate set, regardless of whether it is revealed in a single attempt or across several. By allowing B > 1, we account for these broader, more realistic adversarial capabilities, making our setting more comprehensive and practical. Please note that this threat model with a budget of B>1 was introduced for LLMs by [1].
>
> We have added this explanation in the updated version of the paper.
>
> > In Section 6.2, only two LLaVA model sizes with 7B and 13B parameters are evaluated to draw conclusions on the influence of model size on vulnerabilities. Other model architectures with varying model sizes should be evaluated for a more convincing conclusion. If additional experiments are not possible, a relevant discussion is needed.
>
>
> - We acknowledge that our evaluation is limited by the availability of multimodal LLMs at different scales—currently, LLaVA models are the only ones available in both 7B and 13B parameter sizes. And we are not aware of any other multimodal architectures available in multiple sizes. However, previous studies on LLM scaling offer relevant insights. Our results show that the larger models when edited for deletion are more robust against attacks. This is supported by observations from other LLM-based studies can provide additional insight into the impact of scaling. For example, [4] found that RLHF models become increasingly difficult to red team as they scale, whereas plain LMs, prompted LMs, and LMs with rejection sampling exhibit a flat trend with scale. On the other hand, [5] found no correlation between robustness and model size within certain model families. This suggests that the robustness of the models might be a function of the unlearning method. In our work, we use model editing for targeted deletion and conclude that the robustness of models edited for deletion improves as their size increases.
>
>
>
>
>
>
>
> [4] Ganguli, Deep, et al. "Red teaming language models to reduce harms: Methods, scaling behaviors, and lessons learned." arXiv preprint arXiv:2209.07858 (2022).
>
>
> [5] Mazeika, Mantas, et al. "Harmbench: A standardized evaluation framework for automated red teaming and robust refusal." arXiv preprint arXiv:2402.04249 (2024).
>
>
> > Equation (2) could be introduced earlier, e.g., in Section 3.1 when it is mentioned.
> - We have made this change in the updated version
>
> > Tables 5 and 6 should be placed before the references.
> - We have made this change in the updated version

---

> > ### Comment · Action_Editor_gAKc · 2024-11-15
> > **VQAv2 does not contain only yes/no questions**
> >
> > > We had tried extending our pipeline to the usual VQAv2 dataset. However, these other VQA datasets contain predominantly Yes/No questions, where the concept of deletion of the answer is less meaningful for assessing factual knowledge removal. Therefore, extending the pipeline to such datasets did not serve the core goals of our approach.
> >
> > While I agree that VQAv2 has a lot of yes/no questions, the overall size is much larger than OK VQA, even if removing yes/no questions.
> >
> > I am sure that VQAv2 has a lot of questions in the style shown in Fig 2 or Table 8, so this argument seems somewhat limited.

---

### Review · Reviewer_Fhpe · 2024-09-28

**Summary Of Contributions:**

# This work presents
##  (A) UnLOK-VQA, a benchmark for the evaluation of unlearning in multimodal Large Language Models (MLLM).
1. Prior methods have focussed on creating data for the unlearning of LLM, but UnLOK-VQA extends and introduces a systematic evaluation of unlearning in MLLMs.
2. This dataset evaluates the unlearning methods in three aspects; (a) __Efficacy:__ Measures how well the information is deleted, __Generalization:__ Evaluates if the method is robust to alternate ways of information extraction (eg: paraphrased prompts)  and __Specificity:__ Measures the damage on the benign (unintended) concepts.


## (B) An attack-and-defence framework
1. The authors have chosen 6 unlearning objects as the defence methods to evaluate.
2. They evaluate the robustness of these 6 methods under both the white-box (4) and black-box (3) attacks.
4. The authors evaluate the 6 models under these 7 attacks on the proposed UnLOK-VQA dataset and present their findings.
5. Additionally, they also propose a white-box attack called $PD^2$.

**Audience:**

Yes

**Broader Impact Concerns:**

A broader Impact statement is provided in the manuscript.

**Claims And Evidence:**

No

**Requested Changes:**

## Clarifications (Critical)
1. When generating the _"Rephrase Image - Medium"_ data, the authors mention that a segment of the image (identified with GroundingDINO) is removed. __Question__: What happens to the removed region? Do the authors use any inpainting tools to fill the missing region?
2. Out of curiosity, why is there no white-box attack focusing specifically on the image side? All the chosen white-box attacks are focussing on only the text side. Similar to Table 4, do the authors have any experiments justifying the need to attack only the text side? Is it possible to design an attack on the image side?
3. In section 6.4, aside from the empirical justification for editing modules within the LLM yielding better results over multimodal projectors,  a discussion on why this pattern is observed should help understand the results better.


## Recommendations (Not critical, but will strengthen the work)
1. Figure 2, which provides a holistic overview, is not easy to follow immediately. It takes some time to understand what it tries to convey. Updating the image would help the reader follow the work more easily.
2. There are a few typos / grammatical erors that can be rechecked and corrected.
3. The chronology of the table can be corrected (eg: Table 2 and Table 3).

**Strengths And Weaknesses:**

# Strengths:
1. The proposed dataset provides a systematic way to understand the robustness of the unlearning methods for MLLMs.
2. The attack-and-defend framework binds together multiple unlearning methods for MLLMs and provides a standard benchmark for the community to build further.

# Weaknesses:
## Insufficient Evaluation / Discussion
1. The authors claim that the proposed white-box attack called $PD^2$, is capable of capturing the "deleted" information. __(Q1)__ What is the motivation for this new attack over the PD attack? __(Q2)__ Why is the performance (ASR) of $PD^2$ poorer than almost all the other attacks?
2. Are there any ablation studies to justify the choice of using the 7th and 9th layers for editing of the 7B and 13B MLLMs? Are there any reasons why they work better than other layers?
3. For the scaling experiment (Figure 3): __(Q1)__ Why did the authors choose to evaluate only using the "Fact-Erasure" defence and not using the best defence method from Table 1 (__HP__) ? __(Q2)__ Given the fine-tuning attack is the strongest white-box attack (from Table-1), why didn't the authors use that instead of the HP attack in Figure 3?  __(Q3)__ The authors highlight that "scaling makes models more robust", but the experimental results in section 6.3 and figure 3, do not provide sufficient data points to support the claim.

---

> ### Author Response · Authors · 2024-10-13
> **Response to Reviewer Fhpe - Part 1**
>
> > What is the motivation for this new attack over the PD attack?
>
> - The motivation for introducing the new PD2 attack arises from the need to evaluate whether a more generalized version of the original Probability-Delta (PD) attack [1] can better capture "deleted" information in multimodal models. The PD2 attack introduces a higher-order comparison of the logit distributions, making it more sensitive to subtle changes in the model's behavior.
> The Head Projection attack can be considered a PD attack of order 0, where no difference is computed, and only the probability distributions are considered.
> The PD attack is an order 1 attack, computing the first-order difference between the consecutive distributions.
> The PD2 attack, by contrast, is an order 2 attack, where the difference of differences between the distributions is computed, providing a second-order comparison. This aims to capture more nuanced traces of deleted information that might be overlooked by simpler approaches.
> We have added this motivation in the revised version of the paper.
>
> [1] Patil, Vaidehi, Peter Hase, and Mohit Bansal. "Can Sensitive Information Be Deleted From LLMs? Objectives for Defending Against Extraction Attacks." The Twelfth International Conference on Learning Representations.
>
> > Why is the performance (ASR) of PD2 poorer than almost all the other attacks?
>
>
> The poorer performance (ASR) of PD2 compared to other attacks is an empirical observation. Intuitively, PD2 can be seen as a second-order extension of the Head Projection attack (order 0) and the PD attack (order 1), making it a "difference of differences" attack. While PD attack is better than HP, PD2 does not improve on top of PD1. This is likely because higher-order attacks delve deeper into differences, making the targeted information less apparent.
>
> We have added this reason in the revised version of the paper
>
> > Are there any ablation studies to justify the choice of using the 7th and 9th layers for editing of the 7B and 13B MLLMs? Are there any reasons why they work better than other layers?
>
>
> We empirically tuned the layers, and the selection of the 7th and 9th layers is based on our observations. Please refer to the "Model editing methods" section in Sec 5, where we mention this. While prior works, such as [2], have attempted to localize information using causal tracing and then edit the corresponding weights, a follow-up study [3] demonstrates that localization does not necessarily guide effective editing. This is why we opted to select layers empirically rather than relying on localization.
>
> We have added this explanation in Sec 5 of the revised version of the paper.
>
> [2] Meng, Kevin, et al. "Locating and editing factual associations in GPT." Advances in Neural Information Processing Systems 35 (2022): 17359-17372.
>
>
> [3] Hase, Peter, et al. "Does localization inform editing? surprising differences in causality-based localization vs. knowledge editing in language models." Advances in Neural Information Processing Systems 36 (2024).
>
> For the scaling experiment (Figure 3):
> > Why did the authors choose to evaluate only using the "Fact-Erasure" defence and not using the best defense method from Table 1 (HP) ?
>
>
> We chose to evaluate using the "Fact-Erasure" defense instead of the HP defense because HP is more sensitive to hyperparameter selection. To analyze the effect of scaling, we aimed to keep the evaluation independent of hyperparameter choices.
> We have added this reason in the revised version of the paper.
>
> > Given the fine-tuning attack is the strongest white-box attack (from Table-1), why didn't the authors use that instead of the HP attack in Figure 3?
>
>
> We excluded the fine-tuning attack in Figure 3 because it modifies the model's weights, which could interfere with analyzing the effect of scaling. While unlearning and scaling interact consistently across methods, fine-tuning introduces a confounding factor, making it unsuitable for this evaluation.
> We have added this reason in the revised version of the paper.

---

> ### Author Response · Authors · 2024-10-13
> **Response to Reviewer Fhpe - Part 2**
>
> > The authors highlight that "scaling makes models more robust", but the experimental results in section 6.3 and figure 3, do not provide sufficient data points to support the claim.
>
> We acknowledge that our evaluation is limited by the availability of multimodal LLMs at different scales—currently, LLaVA models are the only ones available in both 7B and 13B parameter sizes. And we are not aware of any other multimodal architectures available in multiple sizes. However, previous studies on LLM scaling offer relevant insights. Our results show that the larger models when edited for deletion are more robust against attacks. This is supported by observations from other LLM-based studies can provide additional insight into the impact of scaling. For example, [4] found that RLHF models become increasingly difficult to red team as they scale, whereas plain LMs, prompted LMs, and LMs with rejection sampling exhibit a flat trend with scale. On the other hand, [5] found no correlation between robustness and model size within certain model families. This suggests that the robustness of the models might be a function of the unlearning method. In our work, we use model editing for targeted deletion and conclude that the robustness of models edited for deletion improves as their size increases.
>
> [4] Ganguli, Deep, et al. "Red teaming language models to reduce harms: Methods, scaling behaviors, and lessons learned." arXiv preprint arXiv:2209.07858 (2022).
>
> [5] Mazeika, Mantas, et al. "Harmbench: A standardized evaluation framework for automated red teaming and robust refusal." arXiv preprint arXiv:2402.04249 (2024).
>
>
>
> > When generating the "Rephrase Image - Medium" data, the authors mention that a segment of the image (identified with GroundingDINO) is removed. Question: What happens to the removed region? Do the authors use any inpainting tools to fill the missing region?
>
>
> The confusion arises from how "replacing an object generated by Grounded SAM" is implemented. In this case, Grounded SAM repaints the part of the image corresponding to the removed object rather than replacing it with a new object. This is mentioned in the last line of the medium rephrase image generation paragraph in Section 3.2. We have clarified this point in the updated version to avoid further misunderstandings.
>
> > Out of curiosity, why is there no white-box attack focusing specifically on the image side? All the chosen white-box attacks are focussing on only the text side. Similar to Table 4, do the authors have any experiments justifying the need to attack only the text side? Is it possible to design an attack on the image side?
>
>
> The whitebox attacks that we choose actually operate on the representations of the LLM. However, the LLMs don’t take only the LLM but also the image tokens. Thus the attacks are on the multimodal representations and cannot be thought of as text-only attacks. Since the information about the question flows directly to the LLM and the image information flows via the ViT into the LLM, it is not possible to elicit information from just fromViT since it has only the image information and lacks the ability to interact with language.
>
>
> > In section 6.4, aside from the empirical justification for editing modules within the LLM yielding better results over multimodal projectors, a discussion on why this pattern is observed should help understand the results better.
>
>
> The improved results from editing within LLM layers, compared to multimodal projectors, stem from the stage of processing. Multimodal projectors operate early, handling raw input translation before the model fully integrates information. In contrast, LLM layers process data later, where final knowledge representations are formed. Editing at this stage is more effective, directly targeting the model's semantic associations, resulting in more precise removal of specific knowledge.
> We have added this discussion in the updated version of the paper.
> > Figure 2, which provides a holistic overview, is not easy to follow immediately. It takes some time to understand what it tries to convey. Updating the image would help the reader follow the work more easily.
>
>
> We have updated the figure in the revised version of the paper.
>
> > There are a few typos / grammatical errors that can be rechecked and corrected.
>
>
> We have addressed the typos and grammatical errors in the revised version of the paper to the best of our ability.
> > The chronology of the table can be corrected (eg: Table 2 and Table 3).
>
>
> The chronology of the tables was reversed to allow for side-by-side placement of Figure 3 and Table 3

---

### Review · Reviewer_ycL5 · 2024-10-04

**Summary Of Contributions:**

This work presents a new dataset that covers the variations on both images and texts, which allows a comprehensive evaluation on multi-modal editing. With a collection of model attack and defense methods, the work demonstrate the value of this dataset, as well as different patterns regarding unimodal vs. multi-modal editing and resilience regarding different model sizes.

**Audience:**

Yes

**Broader Impact Concerns:**

No significant concern

**Claims And Evidence:**

Yes

**Requested Changes:**

- It would be great to have more explanation and justification on the choices in experiment design
- It would be great to have a more in-depth analysis on the data quality, for example, the quality of generated paraphrases.

**Strengths And Weaknesses:**

**Strengths**

- Present a dataset for evaluating multi-modal editing.
- An extensive study on different attack and defense methods that demonstrate the value of this dataset.
- Demonstrate that the discrepancy between different model editing methods and the model behavior regarding different model sizes.

**Weaknesses**

- The definition of easy/medium/hard seems to be arbitrary, how do we know they are actually on that level?
- In addition to the difficulty level, what are the quality of the examples? A in-depth analysis on generated data quality would be great.
- Long description about how to generate examples without much statistics or some information about the dataset, or any information about dataset quality.
- This work did an extensive study on attack and defense methods. However, I found it is difficult to get some insights from this work. In addition to the experiment results that demonstrate multimodal editing is much more effective than unimodal editing, I would like to see more explanation and justification of the experiment design.

---

> ### Author Response · Authors · 2024-10-13
> **Response to Reviewer ycL5 - Part 1**
>
> > The definition of easy/medium/hard seems to be arbitrary, how do we know they are actually on that level?
>
> The definition of "easy," "medium," and "hard" difficulty levels in rephrase and neighborhood samples is by design based on their proximity to the sample that is being deleted and our ablation experiments validate these distinctions.
> For rephrase samples, as shown in Table 2, both easy and medium rephrase images exhibit similar attack success rates against most defenses, while hard rephrase images are significantly more effective in bypassing defenses. Similarly, hard question rephrases—often based on jailbreak prompts—achieve the highest attack success rates compared to easy and medium rephrases. The increased effectiveness of hard rephrase images and questions likely stems from the fact that while their semantic content remains intact, the general composition or phrasing is significantly altered, making it harder for defense mechanisms to detect or mitigate them.
> For neighborhood samples, Table 3 shows that hard neighborhood images result in a higher Image Neighborhood Delta-Acc compared to easy neighborhood images. This is because hard neighborhood images, by design, lie closer to the deleted sample point, making them more challenging to defend against. These ablation results confirm that the intuitive classification of difficulty levels is reflected in the model's susceptibility to attacks.  See Sec 6.3 for detailed explanation on these ablations.
> Figures 5 and 6 further illustrate that rephrase points lie closer to the target data point than neighborhood data points. And neighborhood points lie closer to the target than random points.
>
>
>
>
> > In addition to the difficulty level, what are the quality of the examples? A in-depth analysis on generated data quality would be great. Long description about how to generate examples without much statistics or some information about the dataset, or any information about dataset quality.
>
> **Response A**
>
> - To verify the data quality, we conducted a human evaluation, as described in Section 3.4. We asked graduate student annotators to assess two main criteria: (1) whether the target answer remains consistent for rephrase data, and (2) whether the target answer appropriately changes for neighborhood data.
>
> - Our results show that over 90% of the evaluated samples meet these criteria (as shown in Table 4), supporting the quality of the dataset.
> - The details of the human evaluation process, including the design of the evaluation questions and interface demonstrations, are provided in Appendix C.
> - For a more detailed dataset analysis, including the number of samples and their components, refer to Section 3.5.
> - The pie chart in Figure 4 illustrates the distribution across various categories within the UnLOK-VQA dataset, providing further insights into the dataset's diversity.

---

> ### Author Response · Authors · 2024-10-13
> **Response to Reviewer ycL5 - Part 2**
>
> > This work did an extensive study on attack and defense methods. However, I found it is difficult to get some insights from this work. In addition to the experiment results that demonstrate multimodal editing is much more effective than unimodal editing, I would like to see more explanation and justification of the experiment design.
>
> **Response B**
> - This work aims to answer the question: Can we delete specific information from multimodal models when we do not want them to know or express this information?
> - Although unlearning techniques have been investigated for text-based LLMs, comparable methods for multimodal LLMs are still largely unexplored. The lack of appropriate datasets and a standardized evaluation framework hampers the ability to assess the effectiveness of unlearning in MLLMs.
> - To address this gap, we first created a dataset specifically designed for information deletion. This dataset evaluates deletion methods along three key dimensions:
>
>
>    - Efficacy – How effectively can the method remove the targeted information?
>    - Generalization – Does the method remain robust, even when faced with adversarial attacks?
>    - Specificity – Does the deletion method only affect the targeted information without damaging unrelated data?
>
> - Additionally, we introduced a threat model to evaluate attack-defense scenarios systematically.
> We chose LoRA fine tuning as the model editing method for targeted deletion, and we performed a comprehensive evaluation of the method in the following sections:
>
> - Section 6.1 evaluates efficacy, generalization, and specificity of the deletion method across various attacks and defenses. While LoRA performs well in removing specific information, our experiments reveal that it is not robust against certain adversarial attacks, highlighting the need for robust deletion defenses.
> - Section 6.2 investigates the role of model scaling in improving robustness. Our experiments show that larger models exhibit increased resilience to attacks, suggesting scaling as a potential defense strategy.
> - Section 6.3 focuses on ablation over the different difficulty levels of rephrase and neighborhood data. We test and verify the hypothesis that difficulty levels are correlated with proximity to the data point and observe trends in how attack success rates vary with this difficulty level.
> - Section 6.4 explores whether editing the multimodal projector yields worse results than editing weights in the LLM layers. This comparison provides insights into more efficient ways to perform multimodal knowledge deletion. The improved results from editing within LLM layers, compared to multimodal projectors, could stem from the stage of processing. Multimodal projectors operate early, handling raw input translation before the model fully integrates information. In contrast, LLM layers process data later, where final knowledge representations are formed. Editing at this stage is more effective, directly targeting the model's semantic associations, possibly resulting in more precise removal of specific knowledge.
> These sections collectively demonstrate the rationale behind our experimental design, and we hope this added context clarifies our approach and findings.
>
> > It would be great to have more explanation and justification on the choices in experiment design
> - Please see the response B above
>
> > It would be great to have a more in-depth analysis on the data quality, for example, the quality of generated paraphrases.
> - Please see the response A above

---

### Author Response · Authors · 2024-10-13
**General Response**

We thank the reviewers for their time and thoughtful feedback. We're pleased that the importance of our new dataset for evaluating multimodal deletion has been recognized. We appreciate the positive feedback on the "extensive study of attack and defense methods" and the "patterns in unimodal vs. multimodal editing and model behavior" (Reviewer ycL5). Reviewers highlighted UnLOK-VQA as a key contribution, acknowledging our systematic approach to evaluating unlearning across the dimensions of efficacy, generalization, and specificity (Reviewer Fhpe) and the clarity of our dataset creation pipeline (Reviewer jwQq). The value of our attack-and-defense framework, covering both white-box and black-box settings, was also appreciated (Reviewer jwQq, Reviewer Fhpe, Reviewer ycL5). Lastly, we are glad our insights into the limitations of LoRA fine-tuning in fully deleting knowledge were found insightful (Reviewer kEAH).

We will now address individual questions from each reviewer below. We have also uploaded a revised version of the paper to address most of the reviewers’ concerns with the changes highlighted in red font.

---

### Decision · Action_Editor_gAKc · 2024-11-19

**Recommendation:** Accept as is

**Comment:**

All reviewers are happy with the author's response and how they address it in the revised version.

The paper can be accepted in its current form.

**Audience:**

All reviewers agree that this is relevant for TLMR's audience. I agree with the reviewers on the assessment: The topic, benchmark, and insights are of interest to the TLMR audience.

**Claims And Evidence:**

The paper set out to create a new dataset for Multimodal Attack-Defense Evaluation and benchmark them, including a new attack approach. The paper provides the dataset and an extensive set of experiments benchmarking attack and defense techniques, providing several insights including the difficulty of completely deleting knowledge from this multimodal data.

All reviewers agree in their final recommendation that the claims are supported by evidence.